# ADAM9-Responsive Mesoporous Silica Nanoparticles for Targeted Drug Delivery in Pancreatic Cancer

**DOI:** 10.3390/cancers13133321

**Published:** 2021-07-01

**Authors:** Etienne J. Slapak, Lily Kong, Mouad el Mandili, Rienk Nieuwland, Alexander Kros, Maarten F. Bijlsma, C. Arnold Spek

**Affiliations:** 1Center of Experimental and Molecular Medicine, University of Amsterdam and Cancer Center Amsterdam, Amsterdam UMC, 1105 AZ Amsterdam, The Netherlands; m.elmandili@amsterdamumc.nl (M.e.M.); c.a.spek@amsterdamumc.nl (C.A.S.); 2Laboratory for Experimental Oncology and Radiobiology, University of Amsterdam and Cancer Center Amsterdam, Amsterdam UMC, 1105 AZ Amsterdam, The Netherlands; m.f.bijlsma@amsterdamumc.nl; 3Oncode Institute, 1105 AZ Amsterdam, The Netherlands; 4Tongji School of Pharmacy, Huazhong University of Science and Technology, Wuhan 430030, China; kongl@hust.edu.cn (L.K.); a.kros@chem.leidenuniv.nl (A.K.); 5Laboratory of Experimental Clinical Chemistry, Department of Clinical Chemistry, Amsterdam UMC, Location AMC, 1105 AZ Amsterdam, The Netherlands; r.nieuwland@amsterdamumc.nl; 6Vesicle Observation Center, Amsterdam UMC, Location AMC, 1105 AZ Amsterdam, The Netherlands

**Keywords:** ADAM9, drug delivery, mesoporous silica nanoparticles, MSN, pancreatic cancer, PDAC

## Abstract

**Simple Summary:**

The clinical efficacy of systemic chemotherapy is limited in pancreatic cancer (PDAC) due to toxicity-dependent dose-limitations often leading to premature cessation of therapy. Targeted delivery of chemotherapeutic drugs to cancer cells, without affecting healthy nontumor cells, will largely reduce collateral toxicity. Reductions in collateral toxicity will allow increased drug concentrations to be used, thereby increasing the efficacy of chemotherapy. In the current study, we designed and validated a PDAC-specific protease-dependent drug release system. More specifically, we generated capped mesoporous silica nanoparticles that only release their cargo after proteolytic removal of the cap by PDAC-expressed proteases. We demonstrated the feasibility of protease-mediated targeted drug delivery in PDAC through the release of paclitaxel, resulting in cytotoxicity in cultured PDAC cells.

**Abstract:**

Pancreatic ductal adenocarcinoma (PDAC) has the worst survival rate of all cancers. This poor prognosis results from the lack of efficient systemic treatment regimens, demanding high-dose chemotherapy that causes severe side effects. To overcome dose-dependent toxicities, we explored the efficacy of targeted drug delivery using a protease-dependent drug-release system. To this end, we developed a PDAC-specific drug delivery system based on mesoporous silica nanoparticles (MSN) functionalized with an avidin–biotin gatekeeper system containing a protease linker that is specifically cleaved by tumor cells. Bioinformatic analysis identified ADAM9 as a PDAC-enriched protease, and PDAC cell-derived conditioned medium efficiently cleaved protease linkers containing ADAM9 substrates. Cleavage was PDAC specific as conditioned medium from leukocytes was unable to cleave the ADAM9 substrate. Protease linker-functionalized MSNs were efficiently capped with avidin, and cap removal was confirmed to occur in the presence of PDAC cell-derived ADAM9. Subsequent treatment of PDAC cells in vitro with paclitaxel-loaded MSNs indeed showed high cytotoxicity, whereas no cell death was observed in white blood cell-derived cell lines, confirming efficacy of the nanoparticle-mediated drug delivery system. Taken together, this research introduces a novel ADAM9-responsive, protease-dependent, drug delivery system for PDAC as a promising tool to reduce the cytotoxicity of systemic chemotherapy.

## 1. Introduction

Pancreatic cancer (PDAC) is a devastating disease with the worst outcome of any human cancer. Despite significant improvement in the treatment of cancer in general, limited progress has been made in PDAC and the reported overall 5-year survival rates are typically less than 9% [1]. This poor prognosis is largely due to the fact that chemotherapeutics are ineffective in PDAC, and it is thus of utmost importance to improve the efficacy of treatment. Importantly, the efficacy of systemic chemotherapy is limited due to toxicity-dependent dose limitations and discontinuation of treatment in approximately 60 and 70% of patients receiving gemcitabine or nab-paclitaxel combination therapy, respectively [2].

Targeted delivery of chemotherapeutic drugs to cancer cells, without affecting healthy nontumor cells, is expected to largely reduce systemic toxicity and to increase the efficacy of chemotherapy. Several drug delivery nanocarriers have been described, and mesoporous silica nanoparticles (MSNs) are frequently used [3]. MSNs are particularly interesting as their porous structure enables high cargo-loading capacity. Additional advantages of MSNs include tunable particle and pore sizes, high biocompatibility, the possibility of functionalizing the inner core and outer surface independently from one another, and the possibility of controlled drug release through the use of a gatekeeper system [4,5,6].

Multiple different gatekeeper systems have been described previously, of which protease-mediated drug release systems are of particular interest in the setting of PDAC [7]. This is due to PDAC being characterized by a large desmoplastic reaction that harbors high protease activity. Protease-dependent drug release systems consist of drug-loaded MSNs which are tightly capped by avidin molecules via linker peptides that contain specific protease cleavage sites. Only when the linker peptide is proteolytically cleaved will the avidin cap be removed, and the drugs released. To exploit this approach for targeted delivery of chemotherapeutics in the setting of PDAC, MSNs need to be capped using linker peptides that are only cleaved in the presence of PDAC-specific proteases. Systemic therapy with such capped MSNs engineered to specifically release (chemotherapeutic) drugs in PDAC would thus allow efficient antitumor activity with limited adverse effects on other tissues.

Here, we identify ADAM9 as a PDAC-enriched protease and functionally show that ADAM9 cleavage sites serve as excellent candidate linker peptides. Indeed, avidin- capped MSNs containing the ADAM9 linker are uncapped by PDAC cells, but not by leukocytes. We demonstrate the feasibility of ADAM9-mediated targeted drug delivery in PDAC through the release of paclitaxel, resulting in cytotoxicity in cultured PDAC cells.

## 2. Results

### 2.1. Identification of PDAC-Enriched Proteases

To design a PDAC-specific protease linker, we first aimed to identify proteases with high expression in PDAC. Specifically, we aimed to select proteases that are not intracellular, where they are hidden from the linker-capped nanoparticle by the cancer cell’s plasma membrane as well as the abundant desmoplasia that is characteristic of PDAC. First, we queried the subcellular localization of all known proteases expressed by the human genome (571 proteases in total [8]) and identified 258 extracellular or transmembrane proteases (Figure 1A). Subsequent expression analysis of the 258 included proteases in three large PDAC cell line datasets, containing a total of 101 samples with 51 unique PDAC cell lines, identified 30 shared proteases that were overexpressed (Figure 1B). To limit the number of proteases for follow up experiments, we next queried gene expression levels of the identified candidate proteases in patient-derived PDAC tissues. Eighteen of the 30 shared proteases demonstrated above-average expression in five out of six databases (Figure 1C), and four of these (a disintegrin and metalloprotease 9, ADAM9; a disintegrin and metalloprotease 10, ADAM10; cathepsin B, CTSB; and cathepsin D, CTSD) were expressed, on average, 1.5 times higher in PDAC as compared with adjacent nontumor tissue (Figure 1D). Next, we assessed the expression levels of ADAM9, ADAM10, CTSB, and CTSD in leukocyte subsets and found that only ADAM9 expression was low in all analyzed cells (Figure 1E). Finally, we determined the expression level of ADAM9 in healthy tissue and found that ADAM9 expression levels were relatively low in all organs analyzed (Figure 1F), thereby identifying ADAM9 as an interesting candidate protease to pursue.

### 2.2. An ADAM9 Substrate Is Specifically Cleaved by PDAC Cells

To assess whether ADAM9 could indeed be used as a PDAC-specific protease for our protease-dependent drug release system, we designed a fluorescent ADAM9-substrate (Figure 2A) containing an ADAM9 cleavage site based on the precursor TNFα cleavage site [22]. Incubation with different concentrations of recombinant human ADAM9 resulted in substrate cleavage in a dose-dependent manner (Figure 2B). Of note, a 100-fold higher concentration of thrombin, a protease present in the blood of PDAC patients, was unable to cleave the ADAM9-substrate. To test whether the ADAM9-substrate is also efficiently cleaved by PDAC-derived ADAM9, we incubated the ADAM9-substrate with CM from different human (PANC-1, MIA PaCa-2) and mouse (Panc02, KP) PDAC cell lines. Importantly, all PDAC cell lines cleaved the ADAM9-substrate, peaking at an 11- to 13-fold increase compared with control (Figure 2C). To assess the specificity of the ADAM9 substrate for PDAC cells, and to assess aspecific substrate cleavage by leukocytes, we incubated the ADAM9 substrate with CM derived from human (THP-1, HL60, U937) and mouse (RAW264.7, MHS) cell lines. As expected based on the bioinformatic analysis described above, no cleavage of the ADAM9 substrate occurred in the presence of CM from leukocytes (Figure 2D). Together, these results suggest that the selected ADAM9-substrate is a promising PDAC-specific linker-peptide for our MSN-based drug release system.

### 2.3. Generation and Validation of ADAM9-Responsive Avidin-Capped MSNs

Next, we set out to generate MSNs containing the biotin-coupled ADAM9-substrate peptide linker (Figure 3A). To this end, we employed a sol-gel procedure for the synthesis of amine-modified MSNs [23]. Surface modification with a PEG-N3 spacer was followed by a copper-catalyzed click reaction conjugating the biotin-coupled ADAM9-peptide linker onto the outer surface of the MSNs (denoted ADAM9-MSN), with the biotin moiety facing the periphery, resulting in the generation of ADAM9-responsive MSNs. The mesoporous nature of the nanoparticles was confirmed by transmission electron microscopy (Figure 3B). Via routinely used physiochemical methods, including Fourier-transform infrared (FT-IR) spectroscopy, zeta potential (ZP), and dynamic light scattering (DLS) measurements, we confirmed the successful synthesis and functionalization of the MSNs (Figure 3C–E). As shown, the PEG-N3 surface modification resulted in FT-IR peaks at 1680 cm^−1^ (carbonyl stretch) and 2160 cm^−1^ (azide stretch), of which the latter disappeared following functionalization with the biotin-conjugated ADAM9-peptide linker (Figure 3C). In line, the ZP increased as a result of the functionalization of MSNs with amine from −25.3 ± 0.4 mV to 21.1 ± 1.4 mV, followed by a decrease to 18.9 ± 0.7 mV as a result of subsequent functionalization with COOH-PEG4-N3 (Figure 3D). Functionalization with the biotin-conjugated ADAM9-peptide linker resulted in an increase of ZP to 22.1 ± 0.8 mV. The particle size increased, as expected, along the series MSN < MSN-NH2 < MSN-PEG4-N3 < ADAM9-MSN-biotin, from 163 to 189 nm (Figure 3E). To functionally validate the efficacy of the biotin-conjugated ADAM9-peptide linker surface modification, we incubated ADAM9-MSNs with FITC-conjugated avidin. Subsequent flow cytometry revealed that 97% of the biotin-conjugated ADAM9-MSNs were FITC-positive, whereas nonfunctionalized bare MSNs remained FITC-negative (Figure 3F). Together, these data show that the conjugation of biotin-coupled ADAM9-linker peptides onto the MSNs was successful and that this allows efficient capping of MSNs with avidin.

### 2.4. ADAM9-Dependent Removal of the Avidin Cap

Next, we aimed to assess whether the ADAM9-peptide linker is efficiently cleaved once conjugated to the MSNs, and whether or not the avidin cap interferes with protease-mediated cleavage. To this end, ADAM9-MSNs were capped and subsequently incubated with DMEM, PANC-1 CM, or PBS (negative control). As expected, we observed a strong decrease in FITC-positive MSNs after PANC-1 CM treatment but not in PBS and DMEM conditions (Figure 4A). Quantification of these signals revealed that treatment with PANC-1 CM resulted in a strong decrease in FITC-positive MSNs compared with untreated MSN-FITC, while DMEM had no significant effect on the percentage of FITC-positive MSNs (Figure 4B). These data show that the ADAM9-peptide linker is not only cleaved in solution but also when conjugated to the MSNs, confirming the validity of our ADAM9-responsive, biotin–avidin-capped drug delivery system. Furthermore, these experiments underscore the ability of tumor cell-derived protease activity to cleave the gatekeeper linker.

### 2.5. Drug Release after CAP Removal

To assess whether ADAM9-dependent cap removal from the MSNs indeed results in drug release, we next performed cytotoxicity experiments. To this end, ADAM9-MSNs were loaded with paclitaxel with an encapsulation efficiency of 10%. Paclitaxel-loaded and empty ADAM9-MSNs were subsequently capped with avidin and a cell viability assay with PANC-1 cells was performed. As expected, capped, empty MSNs did not affect cell viability, whereas capped paclitaxel-loaded MSNs induced PANC-1 cell death in a dose-dependent manner (Figure 4C). To determine the specificity of PTX-loaded ADAM9-MSNs for PDAC cells, the white blood cell-derived cell line THP-1 was included in the analysis. The IC50 values of free PTX for cell death in PANC-1 (2.3 nM) and THP-1 (2.3 nM) cells were similar (Figure 4D). Treatment of THP-1 cells with PANC-1 IC50 values of capped PTX-loaded MSNs did not result in cell death. Of note, uncapped PTX-loaded MSNs efficiently killed white blood cells (Figure 4E). Together, these results demonstrate that leakage of PTX from capped MSNs is negligible and that proteolytic removal of the avidin cap is PDAC-specific and results in efficient release of paclitaxel from the ADAM9-MSNs, thereby underscoring its potential as an anticancer modality.

### 2.6. Pan-Cancer Applicability of the ADAM9-Responsive Nanoplatform

ADAM9 overexpression has been reported in several cancer types [24], suggesting that the ADAM9-MSN delivery system may hold promise in other malignancies as well. To validate this hypothesis, we assessed ADAM9 levels in a large set of cancer cell lines. Interestingly, ADAM9 is highly expressed relative to other proteases in most cell types, confirming that ADAM9 expression is indeed upregulated in many cancers (Figure 5A). To functionally confirm that our ADAM9-responsive MSNs could indeed be employed across tumor types, we assessed ADAM9-substrate cleavage using conditioned medium collected from different gastrointestinal cancer cell lines (i.e., bladder, colon, duodenum, and esophageal). All CMs cleaved the ADAM9-substrate rather efficiently, with most cell lines reaching maximum cleavage within 15 min (Figure 5B). These observations strongly suggest that the potential of our ADAM9-based MSNs extends beyond PDAC, and could serve as a targeted delivery system across gastrointestinal cancer types.

## 3. Discussion

Systemic drug delivery in PDAC is relatively ineffective, necessitating high-dose treatment strategies. The adverse effects associated with high-dose therapy are severe, and a large proportion of patients require cytotoxicity-dependent dose limitations, or even treatment discontinuation [2,25]. In the present study, we designed a protease-dependent drug delivery system that specifically targets PDAC cells. Specifically, we generated MSNs that release their cargo (i.e., chemotherapeutics) in the presence of PDAC cells but not healthy leukocytes, suggesting these MSNs may be a promising tool to reduce the cytotoxicity of systemic chemotherapy.

Limiting chemotherapy-dependent cytotoxicity is of obvious clinical importance as more patients would be eligible for treatment and fewer patients would require dose reductions. Reducing toxicity may also affect morbidity. Indeed, many patients that receive FOLFIRINOX or nab-paclitaxel/gemcitabine combination therapy need toxicity-dependent supportive care during treatment [2], consisting of blood transfusions, antiemetics, hydration procedures, and anemia treatments. Of interest, the total cost of supportive care surpasses the cost of first-line treatment [2]. Therefore, reducing side effects through targeted delivery of chemotherapeutics could increase the number of patients eligible for chemotherapy treatment, while simultaneously reducing patient morbidity and the costs of counteracting side effects.

The ADAM9-dependent targeted drug release system described here builds on existing nanoparticle formulations but with increased target specificity. Of note, albumin nanoparticle-bound paclitaxel (nab-paclitaxel) was considered the first targeted delivery method in PDAC. Albumin was originally thought to bind to SPARC (secreted protein acidic and rich in cysteine) in the tumor microenvironment [26], however more recent experimental data revealed that increased delivery and antitumor activity of nab-paclitaxel does not depend on the interaction with SPARC [27,28]. Although nab-paclitaxel does not appear to specifically target PDAC cells, it remains widely used as a first-line treatment of PDAC patients due to its superior safety profile compared with other formulations of paclitaxel [29]. Several alternative targeted delivery systems have been explored over the last years, of which MSNs are of particular interest as their structure enables high drug-loading capacity and specific surface modifications to allow targeted drug release [5]. In the current study, we used a specific protease-dependent gatekeeper system based on the notion that PDAC is characterized by a desmoplastic reaction harboring an abundance of proteases. This approach is expected to confer a specificity previously unattainable in the treatment of PDAC.

In recent years, proteolytic activity-based MSNs have been reported for sarcoma [30], squamous cell [31], colon [31,32], lung [7,33], and breast cancer [34]. All these nanoplatforms employ gatekeeper systems containing matrix metalloproteinase (MMP)-2 and -9 protease linkers, based on the observation that these MMPs are frequently upregulated in cancer [35,36] and the fact that MMP cleavages site are well described. Importantly, instead of opting for an MMP2- or MMP9-dependent gatekeeping system, an unbiased approach was used to identify truly PDAC-enriched proteases, which identified ADAM9 as an attractive candidate protease. Indeed, ADAM9 is overexpressed in PDAC compared with healthy pancreatic patient tissue, while ADAM9 expression levels were low in different healthy blood cell populations and healthy solid organs. Interestingly, our unbiased approach did not identify MMP2 and MMP9 as potential candidate proteases, mainly because their expression levels were low as compared with ADAM9 expression. Of note, MMPs may also lack PDAC specificity as most MMPs are expressed by monocytes [37], whereas MMP2 and MMP9 are also expressed in the bone marrow [38]. This suggests that ADAM9-based gatekeeper systems may be superior to MMP2- and MMP9-based systems for the development of proteolytic activity-based delivery mechanisms. In addition to the potential superiority of ADAM9 over more commonly used proteases, we present here the first proteolytic activity-based MSN for targeted delivery of chemotherapeutics in PDAC. 

After the bioinformatics identification of ADAM9 as a PDAC-enriched protease potentially suitable to be employed in PDAC-targeted drug release, we functionally validated the applicability of ADAM9 using ADAM9-substrate cleavage experiments. We demonstrated that both recombinant and PDAC cell-derived ADAM9 efficiently cleaves a substrate containing a previously described ADAM9 cleavage site present in TNFα [22]. Of note, cleavage was specific as excess amounts of thrombin (a protease and coagulation factor often elevated in PDAC patients [39]) did not result in cleavage. More importantly, incubation with leukocyte-derived medium did not result in any cleavage of the ADAM9 substrate either, confirming the substrate to be PDAC-specific and suggesting that a peptide linker containing the ADAM9 cleavage site would serve as an ideal candidate to cap our MSNs. We confirmed this observation by demonstrating that ADAM9-peptide linker functionalized MSNs are readily uncapped by PDAC cells. This indicates that neither conjugation to MSNs nor avidin capping of the ADAM9 linker peptide leads to steric hindrance that precludes ADAM9-dependent cleavage. Importantly, incubation of PDAC cells with paclitaxel-loaded ADAM9-MSNs induces cytotoxicity, suggesting that linker cleavage and cap removal leads to drug release and that avidin does not block the pores of MSNs once the peptide linker is cleaved. We furthermore confirmed the specificity of our ADAM9-MSNs for PDAC cells by showing that capped paclitaxel-loaded MSNs do not induce cell death in white blood cell-derived cell lines. Overall, we identified ADAM9-MSNs as promising proteolytic candidates for the targeted delivery of chemotherapeutics in PDAC.

A limitation to the applicability of our ADAM9-MSNs, and one that in fact pertains to all current treatments, could be limited penetration and accumulation in the tumor due to the dense stroma present in PDAC. One method to overcome this potential limitation would be to modify the ADAM9-MSNs with iRGD, a cyclic peptide involved in tumor targeting that has been shown to increase tumor invasion in PDAC [40]. Of note, MSNs are especially appropriate for such strategies as they are highly amenable to the addition of moieties. 

Paclitaxel in albumin-bound formulation, in combination with gemcitabine, serves as the first-line treatment regimen for metastatic PDAC and, consequently, we opted to load our MSNs with paclitaxel. Importantly, however, MSN loading is not specific to paclitaxel and all known chemotherapeutics can be loaded with high efficiency [41,42,43]. Of note, loading is not even restricted to chemotherapeutics and a plethora of proteins have already been successfully loaded [44,45]. In essence, any desired payload can be loaded into ADAM9-MSNs which obviously greatly increases their applicability, with potential utility in, for instance, diagnostic purposes. Moreover, ADAM9 is not only enriched in PDAC but seems to be overexpressed in several cancer types [24]; consequently, we hypothesized that our ADAM9-responsive delivery system may be used outside of PDAC. We provide initial proof for this notion by showing that ADAM9 is upregulated in, amongst others, bladder, colon, duodenum, and esophageal cancers. More importantly, CM derived from bladder, duodenum, colorectal, and esophageal cancer cell lines efficiently cleaved the ADAM9-substrate—indeed indicating that our ADAM9-MSN platform might be applicable to multiple types of cancer.

## 4. Materials and Methods

### 4.1. Analysis of Publicly Available Gene Expression Datasets

Datasets were derived from Gene Expression Omnibus (https://www.ncbi.nlm.nih.gov/gds, accessed on 13 April 2020), and analyzed using the R2 microarray analysis and visualization platform (http://r2.amc.nl, accessed on 13 April 2020). Protease expression levels were derived from cell line datasets (GSE36133, GSE57083, and E-MTAB-783), from PDAC datasets (containing both tumor and control pancreatic tissue; GSE17891, GSE16515, GSE32676, GSE15471, GSE62452, and GSE36924), and from datasets of multiple hematopoietic subtypes (GSE24759), leukocytes (GSE22886), lymphocytes (GSE46510), macrophages (GSE2125), monocytes (GSE7158), whole blood (GTeX v4), and healthy organs (GTeX v4). Subcellular localization was determined using the ‘Subcellular Localization’ category of the Gene Ontology project on uniprot.org. The average expression level was derived by averaging the expression level of all proteases (*n* = 258) analyzed in all samples in a given dataset. For pan-cancer differential ADAM9 expression analysis, the same cell line datasets GSE36133, GSE57083, and E-MTAB-783 were used. Expression levels were considered to be high when they were above the average expression level plus 1.5 × the standard deviation of all proteases. Of note, all comparative analyses (except those using healthy organ datasets) were performed within one dataset to exclude effects from different gene expression analysis platforms and normalization methods. The comparisons between healthy organs and PDAC, as depicted in Figure 1F, used different gene expression analysis platforms and the data were, consequently, first normalized to b-actin expression.

### 4.2. Cell lines, Tissue Culture, and Conditioned Medium Collection

Human PANC-1 and MIA PaCa-2 PDAC (ATCC, Manassas, VA, USA), murine KP (derived from pancreatic adenocarcinomas from p48-Cre/LSL-Kras/Tp53flox/flox mice, kindly provided by Dr. DeNardo, Washington University Medical School, St. Louis, MO, USA), and Panc02 (kindly provided by Dr. Schmitz, Universitätsklinikum Bonn, Bonn, Germany) PDAC cell lines were cultured in DMEM (Lonza, Basel, Switzerland). The human THP-1, U-937, HL-60, and murine RAW264.7 and MH-S white blood cells (all ATCC) were cultured in RPMI. HuTu 80 and MDST8 cells were kindly provided by the Sanger Institute (Cambridge, UK) and grown in DMEM/F12. The OE19, OE33, and FLO-1 (all ATCC) cell lines were grown in RPMI, whereas T24 (ATCC) and RT-112 cells were grown in McCoy’s 5A modified medium (Sigma-Aldrich, Saint-Louis, MO, USA) and EMEM (Lonza,), respectively. The patient-derived cell lines AMC-EAC-081R and AMC-EAC-289B were established in our lab, as previously described in [46], and grown in DMEM. All cell lines were supplemented with 10% fetal calf serum (FCS), 2 mM L-glutamine, 100 units/mL penicillin, and 500 μg/mL streptomycin (all Lonza). Cells were maintained in a humidified incubator at 37 °C and 5% CO_2_. Conditioned medium (CM) was collected by growing cells to 50% confluency in a T75 flask, after which the growth medium was refreshed and collected after 96 h. All cell lines were tested for the absence of mycoplasma on a monthly basis.

### 4.3. Peptide Cleavage Assay

A Dabcyl- and FAM-modified linker peptide containing the ADAM9 cleavage site Ser-Pro-Leu-Ala-Gln-**Ala-Val**-Arg-Ser-Ser-Lys (SPLAQ**AV**RSSK; ADAM9-peptide linker, cleavage sequence shown in bold [22]) was purchased from GL Biochem (Shanghai, China). The peptide (1 μM) was incubated in assay buffer (50 mM Tris-HCl, 20 mM NaCl, 2 mM CuCl_2_, 10 μM ZnCl_2_ in PBS) supplemented with thrombin (0.5, 1, or 2 units/mL; Sigma-Aldrich), ADAM9 (0.02, 0.01, or 0.005 units/mL; R&D Systems, Minneapolis, MN, USA) or CM from PANC-1, MIA PaCa-2, Panc02, KP, THP-1, RAW264.7, U-937, HL-60, or MH-S cells. Additional peptide cleavage assays were performed with CM from HuTu-80, MDST8, OE19, OE33, FLO-1, T24, RT-112, AMC-EAC-081R, and AMC-EAC-289B. Fluorescence was measured every 15 min at Ex/Em 485/528 nm wavelengths using a Biotek Synergy HT plate reader (Biotek Instruments, Winooski, VT, USA).

### 4.4. Synthesis of MSNs

MSNs were synthesized using the sol-gel emulsion procedure from tetraethyl orthosilicate (TEOS, ≥99%, Sigma-Aldrich) with cetyltrimethylammonium bromide (CTAB, Sigma-Aldrich, ≥99%) as surfactant for the mesoporous channels, as previously described in [47]. In short, 2.0 g (5.4 mmol) CTAB was dissolved in 960 mL of deionized water and 7 mL of 2 M NaOH and heated to 80 °C for 30 min, after which 9.0 g (43.2 mmol) tetraethyl orthosilicate (TEOS, ≥99%, Sigma-Aldrich) was added. After two hours incubation at 70 °C, the surfactant was removed by overnight incubation in a methanol (310 mL, >99.8%, Riedel-de-Haën, Seelze, Germany) and hydrochloric acid (31 mL, 37%, Fluka, Buchs, Switzerland) mixture under an inert atmosphere. The resulting particles were filtered over a Buchner funnel, washed several times with ethanol and water, and dried under high vacuum to produce a white solid. Surface functionalization was achieved through overnight incubation in 10 mL dried toluene with 22.1 mg aminopropyl triethoxysilane (APTES, 0.1 mmol, ≥99%, Sigma-Aldrich) in an inert atmosphere. Finally, the mixture was filtered using a Buchner funnel with filter paper and dried under vacuum to obtain MSN-NH2.

Successful synthesis of MSN-NH2 was confirmed by transmission electron microscopy (TEM). TEM samples were prepared by placing a 10 µL droplet of MSN suspension on a 200-mesh copper grid. After 1 min, excess liquid was removed by manually blotting with filter paper, followed by TEM using a JEM-1400 Plus (JEOL) transmission electron microscope operated at 80 kV and equipped with a CCD camera.

### 4.5. Surface Modification of the MSNs

Modification of the MSNs with PEG_4_-N_3_ was performed (as described in [23]) to inhibit interactions between the peptide and the MSN, thereby protecting the conformation of the peptide. In short, stirring 291.3 mg 15-Azido-4,7,10,13-tetraoxa-pentadecanoic acid (N_3_-PEG_4_-COOH, 1 mmol, >98%, Iris Biotech, Marktredwitz, Germany) with 140 mg *N*-hydroxysuccinimide (NHS, 1.2 mmol, ≥98%, Alfa Aesar, Haverhill, MA, USA) in 1 mL dimethylsulfoxide (DMSO, Biosolve, Valkenswaard, The Netherlands) containing 400 mg *N*-(3-(dimethylamino)propyl)-*N*’-ethylcarbodiimide hydrochloride (EDC, 2.1 mmol, Sigma-Aldrich) and 160 mg hydroxybenzotriazole (HOBt, 1.2 mmol, Biosolve) for 30 min under N_2_ gas atmosphere in an ice bath resulted in the *N*-hydroxysuccinimide ester of N_3_-PEG_4_-COOH (NHS-1). Next, 100 mg of MSN-NH_2_ in 4 mL DMSO was added to the NHS-1 reaction mixture and stirred under N_2_ gas atmosphere for 72 h at room temperature, after which the mixture was filtered and washed with deionized water and ethanol (96%) to obtain MSN-PEG_4_-N_3_.

### 4.6. Synthesis of Linker Peptides

Peptides were synthesized with standard solid phase peptide synthesis protocols using a Liberty Blue automated, microwave-assisted, peptide synthesizer (CEM Corporation, Matthews, NC, USA). Peptides were prepared on a 0.1 mmol scale using Tentagel S RAM resin (Rapp Polymere, Tubingen, Germany) with a loading capacity of 0.2–0.27 mmol/g. Fmoc deprotection was performed using 20% piperidine (Biosolve) in dimethylformamide (DMF, Biosolve) at 90 °C for 60 seconds. Amide coupling was accomplished using protected amino acid (5 eq.), the activator diisopropylcarbodiimide (DIC, 5 eq., Sigma-Aldrich), and the activator base Oxyma Pure (5 eq., Carl Roth, Karlsruhe, Germany), heated at 95 °C for 240 seconds. All protected amino acids were purchased from Novabiochem (Merck, Darmstadt, Germany) except for Fmoc-propargyl-Gly-OH, which was purchased from Glentham Life Sciences (Corsham, UK). Subsequent biotinylation of the peptide N-terminus was performed using biotin (3 eq., Sigma-Aldrich, ≥99%, lyophilized powder), chlorobenzotriazole tetramethyluronium hexafluorophosphate (HCTU, 3eq., Novabiochem), and di-isopropylethylamine (DIPEA, 6 eq., Carl Roth).

Peptides were cleaved from the resin for 1 h using a mixture of trifluoroacetic acid (Biosolve), tri-isopropylsilane (Sigma-Aldrich), and water (TFA:TIPS:H_2_O 95:2.5:2.5 *v*/*v*/*v*), followed by precipitation in cold diethyl ether (Sigma-Aldrich). The peptides were then collected by centrifugation, dried under nitrogen flow, and purified by HPLC on a Shimadzu system (Kyoto, Japan) consisting of two KC-20AR pumps and an SPD-20A or SPD-M20A detector equipped with a Kinetix Evo C18 column. Eluents consisted of 0.1% TFA in water (A) and 0.1% TFA in acetonitrile (B), with all peptides eluted using a gradient of 20–90% B over 15 min, with a flow rate of 12 mL/min. Collected fractions were checked for purity via LCMS, with the pure fractions being pooled and lyophilized. LCMS spectra were recorded using a Thermo Scientific TSQ quantum access MAX mass detector connected to an Ultimate 3000 liquid chromatography system fitted with a 50 × 4.6 mm Phenomenex Gemini 3 μm C18 column. The LCMS spectrum of the purified peptide can be found in the supporting information.

### 4.7. Conjugation of MSNs with Linker Peptides

To synthesize linker-modified MSNs, 2.16 mg copper(II)sulfate anhydrous (CuSO_4_, 14.5 µmol, ≥98%, Acros Organics (Thermo Fischer, Waltham, MA, USA)) and 12.5 mg sodium L-ascorbate (61.6 µmol, ≥99%, Sigma Aldrich) were dissolved in 500 µL deionized water and 3 mg tris(3-hydroxypropyltriazolylmethyl)amine (THPTA, 8.7 µmol, Iris Biotech) in 250 µL DMSO was added to the reaction mixture and flushed with nitrogen for 30 min to remove oxygen. Next, 15.8 mg biotin-coupled ADAM9-linker peptide (biotin-SPLAQAVRSKGGG-propargyl; 10 µmol) and 10 mg MSN-PEG_4_-N_3_ were added and stirred overnight at room temperature yielding ADAM9-linker peptide-coupled MSNs (ADAM9-MSN).

### 4.8. Characterization of MSNs

Fourier-transform infrared spectroscopy (FT-IR), zeta potential (ZP), and dynamic light scattering (DLS) measurements were performed to confirm the synthesis of ADAM9-MSN. The FT-IR absorbance of 0.5 mg dry MSNs was measured using a Spectrum Two FT-IR Spectrometer (PerkinElmer, Waltham, MA, USA) over the spectral range of 450–4000 cm^−1^. To determine the ZP, 1 mL MSN solution with a salt concentration lower than 20 mM in universal dip cuvettes was measured, in triplicate, at 25 °C on a Zetasizer Nano-7 S (Malvern Instruments, Malvern, UK) equipped with a 633 nm laser wavelength and a 173° fixed scattering angle. The hydrodynamic diameter was determined by DLS measurements. For this, 1 mg of MSNs was dissolved in 1 mL of 10 mM PBS by sonication, after which 500 µL was measured in triplicate at 25 °C in a plastic cuvette using a Zetasizer Nano-7 S (Malvern Instruments) with a 633 nm laser wavelength and a 173° fixed scattering angle.

### 4.9. Flow Cytometric Analysis of Capping Efficiency

Three milligrams of ADAM9-MSN was dissolved in 1.24 mL FITC-conjugated avidin (Invitrogen, Carlsbad, CA, USA, cat#434411), thoroughly vortexed, and sonicated for 2 min. The mixture was then incubated on a tube rotator in the dark at room temperature. After one hour, the suspension was centrifuged at 14,000 rpm for 5 min and the supernatant was discarded. The MSNs were washed 5 times with 1 mL PBS, after which the FITC-avidin-capped MSNs (ADAM9-MSN-FITC) were dissolved in 600 μL PBS. Subsequently, 40 μL (=200 µg) of ADAM9-MSN-FITC was added to 1 mL of DMEM or PANC CM, or to 80 μL of thrombin or ADAM9 mixture, with a final concentration of 2/1/0.5 U/mL and 0.02/0.01/0.005 U/mL, respectively. The MSNs were incubated for 16 h on a tube rotator in the dark at room temperature. Next, the MSNs were washed twice, dissolved in 400 μL sterile PBS (Gibco, Thermo Fischer), and analyzed on an Apogee A60 Micro-PLUS (Apogee, Hemel Hempstead, Hertfordshire, UK). The resulting data were calibrated and normalized to control beads with known size and fluorescence intensity and subsequently analyzed by Falcon software (Seattle, WA, USA). Falcon software is custom-built software specifically developed for automated data calibration and processing.

### 4.10. Paclitaxel Loading of ADAM9-MSNs

ADAM9-MSNs were loaded with paclitaxel by the adsorption equilibrium method [48]. In short, one milligram of ADAM9-MSN was immersed in 1.75 mL dichloromethane (DCM, AnalaR NORMAPUR, VWR, Radnor, PA, USA) containing 10 mg/mL paclitaxel (LC laboratories, Woburn, Canada). The suspension was vortexed, sonicated in an ice bath for 20 min, vortexed again, and finally sonicated for an additional 10 min in an ice bath. The suspension was then incubated in a thermomixer at 37 °C and 1200 rpm for 1.5 h. Loaded ADAM9-MSNs were centrifuged and washed three times with DCM. To determine the entrapment efficiency of paclitaxel, the loading and wash solutions were dried and resolved in 1.75 mL DMSO and UV absorbance was measured at 274 nm using a Cary 300 UV-Vis spectrometer (Agilent Technologies, Santa Clara, CA, USA). The entrapment efficiency of paclitaxel ranged from 81 to 90% and was calculated as follows: Entrapment efficiency = loaded paclitaxel/initially added paclitaxel.

### 4.11. Avidin Capping of MSNs

Paclitaxel-loaded or control ADAM9-MSNs (1 mg) were dried and dissolved in 500 µL of HBSS (Gibco, Thermo Fischer), tip-sonicated three times for 5 seconds at 20% amplitude, and added to 500 µL of HBSS buffer containing 1.5 mg avidin (EMD Millipore, Burlington, MA, USA). The solution was vortexed for 20 seconds and left for 30 min under static conditions at room temperature. The avidin-capped ADAM9-MSNs were centrifuged (2200 rcf, 3 min, 15 °C) and washed three times with HBSS buffer. The particles were dispersed in 1 mL of HBSS and used for in vitro studies.

### 4.12. MSN-Induced Cytotoxicity

PANC-1 cells (3500) and THP-1 cells (4000) were seeded in a 96-well plate and incubated with different concentrations of capped paclitaxel-loaded or empty ADAM9-MSNs, uncapped paclitaxel-loaded ADAM9-MSNs, or free paclitaxel in a 2:1 ratio of CM and DMEM. After 72 h, PANC-1 cells were washed and incubated with crystal violet (3% formaldehyde, 0.5% crystal violet in dH_2_O) at room temperature. After 20 min, the crystal violet solution was removed, cells were washed 3 times with tap water, and 100 µL/well DMSO was added to solubilize the formed crystals. After 20 min of incubation on a plate shaker at room temperature, the absorbance was measured at 600 nm on a Synergy HT plate reader. After 72 h, THP-1 cells were incubated with 20 µL CellTiter-Blue (Promega, Leiden, The Netherlands) at 37 °C for 3 h. After 3 h fluorescence was measured at 590 nm on a Synergy HT plate reader.

## 5. Conclusions

In conclusion, we designed an ADAM9-responsive drug delivery system that allows drugs of choice to be delivered to cancer cells. Delivering drugs to cancer cells, without affecting healthy nontumor cells, will largely reduce collateral toxicity and allow increased local drug concentrations to be used, thereby increasing the efficacy of therapy.

## Figures and Tables

**Figure 1 cancers-13-03321-f001:**
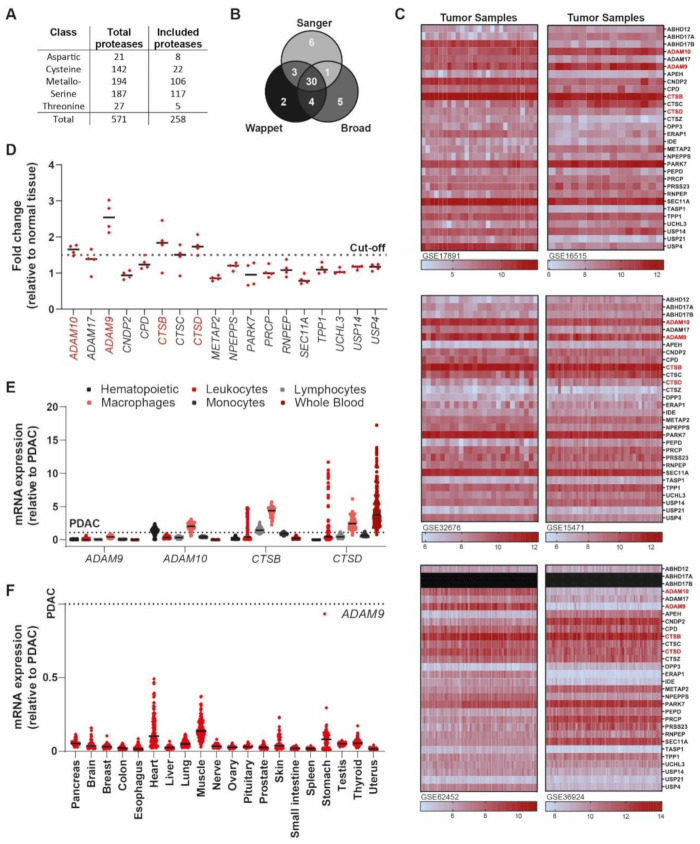
Identification of ADAM9 as a PDAC-enriched extracellular protease. (**A**) Overview of the total number of proteases in the genome and the number of extracellular proteases included in our analysis. (**B**) Venn diagram of the proteases with above-average expression in every PDAC cell line present in the Sanger (E-MTAB-783 [9]), Wappet (GSE57083), and Broad (GSE36133 [10]) datasets. (**C**) Heatmaps showing the expression of 30 upregulated proteases in six datasets involving 268 patients in total (GSE17891 [11], GSE16515 [12], GSE32676 [13], GSE15471 [14], GSE62452 [15], and GSE36924 [16] PDAC datasets). (**D**) Fold-change in expression levels of candidate proteases in patient-derived tumor biopsies, as compared with healthy adjacent pancreas sections. Indicated is the cutoff, set at a 1.5-fold increase. (**E**) Expression levels of selected candidate proteases in healthy blood components, relative to their expression levels in PDAC. Results of hematopoietic subtypes (GSE24759 [17]), leukocytes (GSE22886 [18]), lymphocytes (GSE46510 [19]), macrophages (GSE2125 [20]), monocytes (GSE7158), and whole blood (GTeX v4 [21]) datasets are shown. (**F**) Expression levels of selected candidate proteases in healthy organs, relative to their expression level in PDAC (GTeX v4). The dotted line represents the expression level in PDAC, set to 1.

**Figure 2 cancers-13-03321-f002:**
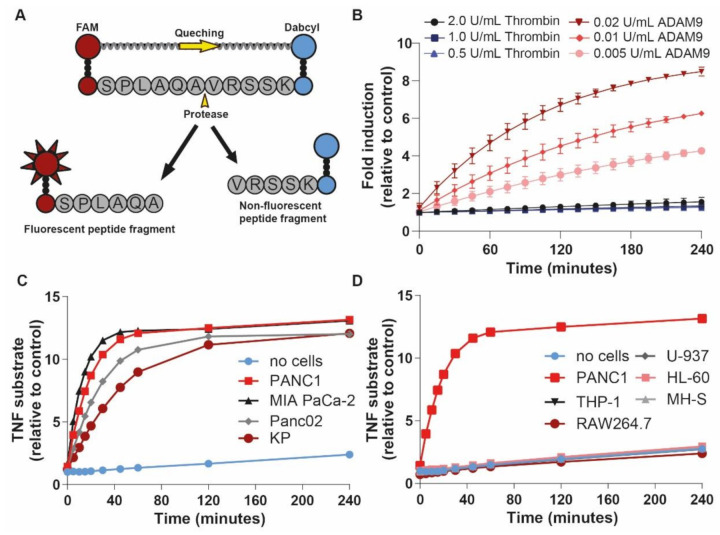
PDAC cells, but not leukocyte subsets, cleave ADAM9 substrates. (**A**) Schematic overview of ADAM9 substrate containing a 5’ fluorophore (FAM) and a 3’quencher (Dabcyl). (**B**) ADAM9 substrate cleavage by recombinant ADAM9 and thrombin (negative control). (**C**) ADAM9 substrate cleavage by CM derived from PDAC cells. (**D**) Lack of ADAM9 substrate cleavage by leukocyte-derived CM (with PANC1 CM as positive control).

**Figure 3 cancers-13-03321-f003:**
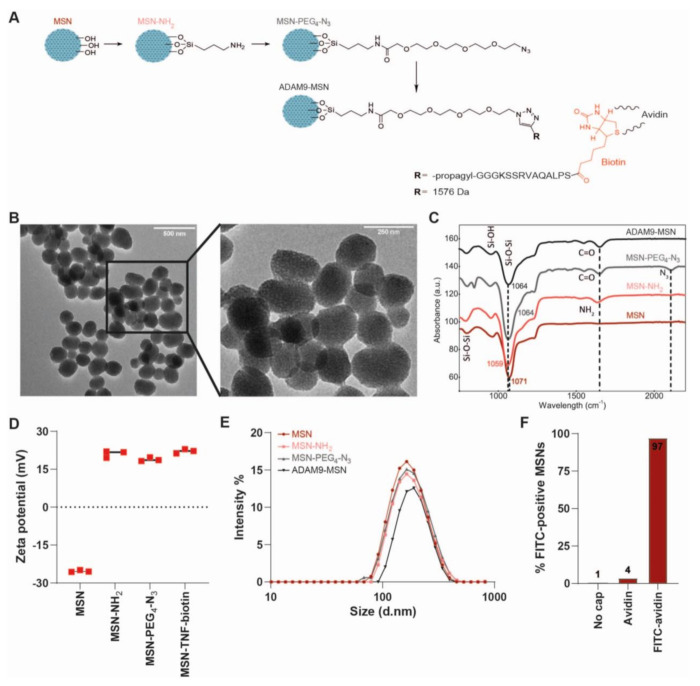
Synthesis and characterization of ADAM9-MSNs. (**A**) Schematic overview of the stepwise synthesis of peptide–biotin coupled MSNs. (**B**) Transmission electron microscopy images of peptide–biotin coupled MSNs. (**C**) Fourier-transform infrared spectra, (**D**) zeta potential, and (**E**) hydrodynamic diameter of the (un)conjugated MSNs. (**F**) Flow cytometry data of FITC-avidin-capped MSNs.

**Figure 4 cancers-13-03321-f004:**
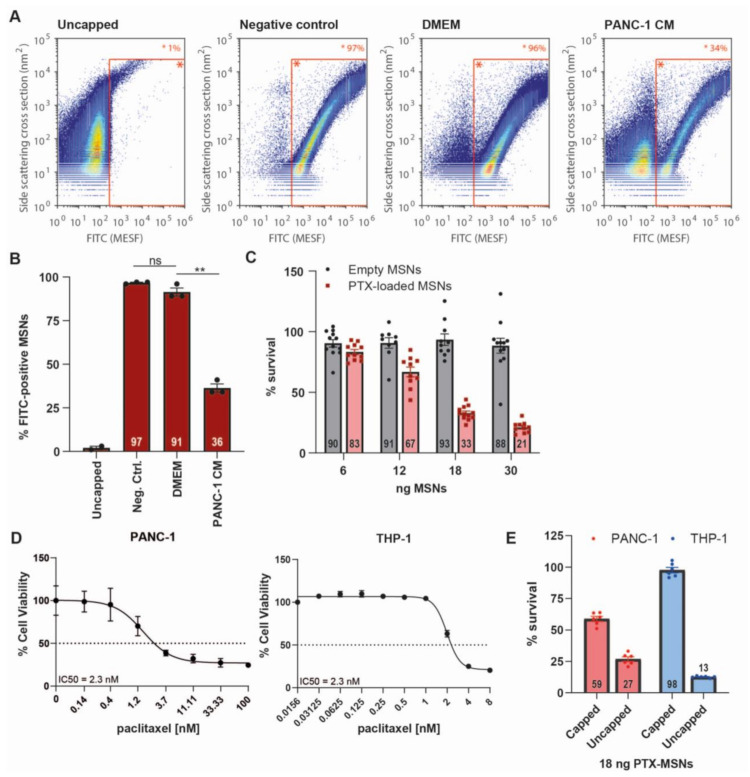
Removal of the avidin cap from MSNs is PDAC-specific and induces cell death. (**A**) Falcon Software-analyzed flow cytometry data showing the MSN distributions of representative samples (*n* = 3) after capping with FITC-avidin and subsequent treatment with DMEM or PANC-1-derived conditioned medium; *x*-axis, FITC (MESH); *y*-axis, side scattering cross section (nm^2^). (**B**) Visualization of percentage of MSNs that are FITC-positive, derived from population plots shown in Figure 4A (*n* = 3, *n* = 2 for uncapped MSNs). ** = *p* < 0.01 as determined by the paired *t*-test. (**C**) Cytotoxicity of avidin-capped, empty and paclitaxel-loaded MSNs on PANC-1 cells after 72 h treatment in vitro. Data are shown as the mean of two experiments with *n* = 6. (**D**) IC50 curves of free paclitaxel on PANC-1 and THP-1 cells after 72 h treatment. (**E**) Cytotoxicity of capped and uncapped paclitaxel-loaded MSNs on PANC-1 and THP-1 cells after 72 h of treatment. Data are shown as the mean of one experiment with *n* = 6. All results are normalized to untreated controls.

**Figure 5 cancers-13-03321-f005:**
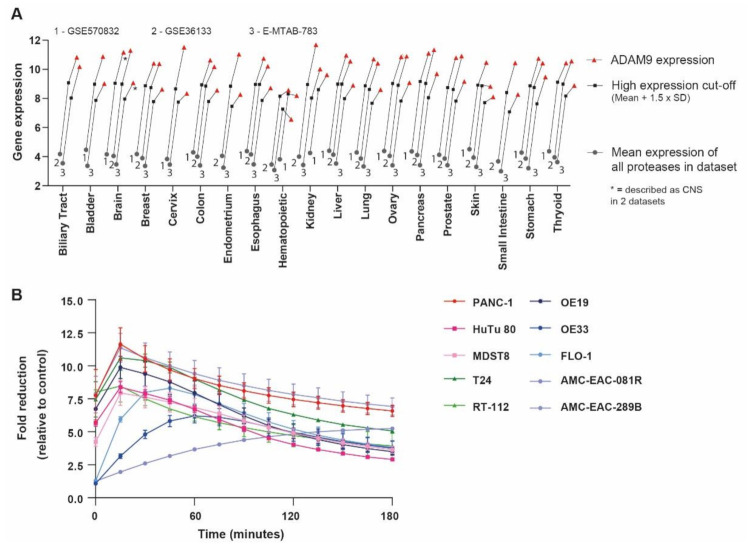
ADAM9-MSNs may be applied to multiple cancer types. (**A**) Average ADAM9 expression in multiple cell lines of several cancer types. Gray circles represent the calculated average expression of all proteases per organ in the large sets of cancer cell lines (GSE36133, GSE57083, and E-MTAB-783); black squares represent the ‘high expression’ cut-off, determined as average protease expression × 1.5 times the standard deviation; red triangles indicate the ADAM9 expression levels. (**B**) ADAM9-substrate cleavage by conditioned media collected from several cancer cell lines. Data are the mean of three experiments performed in duplicate.

## Data Availability

The data presented in this study are available from the corresponding author upon reasonable request.

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
