# Peer review of "ADAM9-Responsive Mesoporous Silica Nanoparticles for Targeted Drug Delivery in Pancreatic Cancer"

_cancers, 2021, doi:10.3390/cancers13133321_

Round 1
Reviewer 1 Report
The manuscript titled "ADAM9-Mesoporous Silica Nanoparticles for targeted drug delivery in Pancreatic Cancer" by Slapak EJ et al., describes a protease dependent drug release of Paclitaxel to be used to treat Pancreatic Ductal Adenocarcinoma (PDAC). This drug delivery system is not completely innovative, as the Paclitaxel formulation is already largely used as a first line intervention to treat many different malignancies. The innovation resides in the final construct, that is made of Mesoporous silica NPs with a capping system able to release Paclitaxel (PTX) only in the presence of ADAM-9, a protease largely found in the microenvironment of PDAC.
In this respect I have found one major concern. Figure 4 represents the key figure of the entire work, as it is where the authors show the efficacy and specificity of their approach. In panel C of that figure they show a dose dependent increase of cell mortality using PTX-loaded MSNs that they compare with the empty MSNs that show non significant toxicity. In order to complete this set of results, two things are missing:
a) it should be also shown the result of the same approach on cell line not expressing ADAM-9, but treated with PTX-loaded and empty MSNs. This experiment is necessary to test the possible leakage of Paclitaxel due to aspecific proteolytic cleavage mediated by other Metalloproteases.
b) Considering a loading efficacy of 90% , It should be also added in figure 4 panel C a third data set where PTX is used as it is, in dose dependent treatment to test the difference in cell viability when the Paclitaxel is available free. This experiment is necessary to test the efficacy of the drug release.
Author Response
Reviewer 1
Figure 4 represents the key figure of the entire work, as it is where the authors show the efficacy and specificity of their approach. In panel C of that figure they show a dose dependent increase of cell mortality using PTX-loaded MSNs that they compare with the empty MSNs that show non-significant toxicity. In order to complete this set of results, two things are missing:
a) it should be also shown the result of the same approach on cell line not expressing ADAM-9, but treated with PTX-loaded and empty MSNs. This experiment is necessary to test the possible leakage of Paclitaxel due to aspecific proteolytic cleavage mediated by other Metalloproteases.
This is a good point and we have addressed this by treating a white blood cell-derived cell line (THP-1; expressing low ADAM9 levels as compared to PDAC cell lines) with capped and uncapped paclitaxel-loaded MSNs. As shown in Figure 4E of the revised manuscript, administration of capped paclitaxel-loaded MSNs at a concentration around the IC50 in PANC-1, did not induce cell death in the white blood cell line. Importantly, uncapped paclitaxel-loaded MSNs very efficiently induced cell death in white blood cells. As outlined in the Results section of the revised version of our manuscript, these data demonstrate that leakage of paclitaxel and aspecific cleavage of the avidin cap by other metalloproteases upon capping of paclitaxel-loaded MSNs is neglectable.
b) Considering a loading efficacy of 90% , It should be also added in figure 4 panel C a third data set where PTX is used as it is, in dose dependent treatment to test the difference in cell viability when the Paclitaxel is available free. This experiment is necessary to test the efficacy of the drug release.
We included free paclitaxel dose response experiments to determine the IC50 values for PANC-1 and THP-1 cells. As shown in Figure 4D of the revised version of our manuscript, IC50 values are very similar, indicating that the absence of cell death in THP-1 after treatment with paclitaxel-loaded MSNs as shown in Figure 4E is not caused by a decreased sensitivity for paclitaxel but by their inability to remove the avidin cap.
Reviewer 2 Report
A Review of cancers-1258382,
Title: ADAM9-Responsive Mesoporous Silica Nanoparticles for Targeted Drug Delivery in Pancreatic Cancer
avidin-biotin gatekeeper system containing a protease linker that is specifically cleaved by tumor cells.
- Graphical abstract: This does not contain important naming and indication for each step. Please add proper captions to effectively summarize the sequential steps of fabrication and treatment.
- Figure 3: (1) Although a chemical structure of ADAM9-peptide linker was shown, it would be more informative to provide peptide sequences along with a length (kDa) for this conjugation. (2) Please also emphasize the technical necessity of PEG-N3 spacer. (3) Isn’t there any additional NMR analysis to support FT-IR data in Figure 3C in order to confirm a proper conjugation?
- Please describe the principle (such as physical entrapment in porous structures in silica particles) for paclitaxel loading into ADAM9-MSN.
Author Response
Reviewer 2
1. Graphical abstract: This does not contain important naming and indication for each step. Please add proper captions to effectively summarize the sequential steps of fabrication and treatment.
We modified the graphical abstract, added the requested additional information, and apologize if this figure was insufficiently self-explanatory.
2.Figure 3: a) Although a chemical structure of ADAM9-peptide linker was shown, it would be more informative to provide peptide sequences along with a length (kDa) for this conjugation.
This is a good suggestion and we modified the figure accordingly (see new Figure 3A in the revised version of the manuscript).
b) Please also emphasize the technical necessity of PEG-N3 spacer.
Again, a good suggestion by the reviewer and we included the rationale for the PEG-N3 spacer to the revised version of our manuscript. Specifically, we included the statement “Modification of the MSNs with PEG4-N3 was performed (as described before [10]) to inhibit interactions between the peptide and the MSN thereby, protecting the conformation of the peptide” on page 14, lines 411–413.
(c) Isn’t there any additional NMR analysis to support FT-IR data in Figure 3C in order to confirm a proper conjugation?
We do not have NMR analysis to support our FT-IR data but feel confident that conjugation was successful based on the functional data showing cell type specific uncapping and drug release. However, we will keep it in mind for future characterizations.
3. Please describe the principle (such as physical entrapment in porous structures in silica particles) for paclitaxel loading into ADAM9-MSN.
We modified the Materials and Methods section to indicate that ADAM9-MSNs were loaded with paclitaxel by the adsorption equilibrium method (see page 16, line 492).